# Synthesizing a $\nu=2/3$ fractional quantum Hall effect edge state from counter-propagating $\nu=1$ and $\nu=1/3$ states

Yonatan Cohen[1,5], Yuval Ronen[1,2,5], Wenmin Yang[1,5], Daniel Banitt[1], Jinhong Park[1], Moty Heiblum[1], Alexander D. Mirlin [3,4], Yuval Gefen[1] & Vladimir Umansky[1]

Topological edge-reconstruction occurs in hole-conjugate states of the fractional quantum Hall effect. The frequently studied filling factor, $\nu = 2/3$, was originally proposed to harbor two counter-propagating modes: a downstream $v = 1$ and an upstream $v = 1/3$. However, charge equilibration between these two modes always led to an observed downstream $v = 2/3$ charge mode accompanied by an upstream neutral mode. Here, we present an approach to synthetize a $v = 2/3$ edge mode from its basic counter-propagating charged constituents, allowing a controlled equilibration between the two counter-propagating charge modes. This platform is based on a carefully designed double-quantum-well, which hosts two populated electronic sub-bands (lower and upper), with corresponding filling factors, $v_l$ and $v_u$. By separating the 2D plane to two gated intersecting halves, each with different fillings, counter-propagating chiral modes can be formed along the intersection line. Equilibration between these modes can be controlled with the top gates' voltage and the magnetic field.

[1] Braun Center for Submicron Research, Department of Condensed Matter Physics, Weizmann Institute of Science, Rehovot 76100, Israel. [2] Department of Physics, Harvard University, Cambridge, MA 02138, USA. [3] Institut für Nanotechnologie, Karlsruhe Institute of Technology, 76021 Karlsruhe, Germany. [4] L.D. Landau Institute for Theoretical Physics RAS, Moscow 119334, Russia. [5] These authors contributed equally: Yonatan Cohen, Yuval Ronen, Wenmin Yang. Correspondence and requests for materials should be addressed to M.H. (email: moty.heiblum@weizmann.ac.il)

In the quantum Hall effect (QHE) regime, charge propagation takes place via downstream chiral edge modes while the bulk is insulating. In the integer QHE (IQHE), the number of downstream edge modes is equal to the number of occupied "spin-split" Landau levels (LLs); each contributes a single edge mode. On the other hand, in the fractional QHE (FQHE) regime, where electron–electron interaction plays a crucial role, the edge profile can be much richer, hosting downstream as well as upstream chiral edge modes[1-3].

It was predicted, nearly 30 years ago[4,5], that the edge structure of the so-called "hole-conjugate" states in the FQHE regime ($i + 1/3 < \nu < i + 1$, where $i$ is an integer and $\nu$ the fractional filling factor) should host counter-propagating modes. The most studied is the $\nu = 2/3$ state, with a downstream $\nu = 1$ mode and an upstream $\nu = 1/3$ mode (Fig. 1a). In the absence of coupling between these modes, this edge structure should yield a "two-terminal" conductance of $G_{2T} = 4e^2/3h$ (Fig. 1b). In practice, however, the measured conductance is always $G_{2T} = 2e^2/3h$—supporting a single downstream $\nu = 2/3$ charge mode and an upstream neutral mode. The experimental ubiquity of the latter conductance value becomes even more remarkable if one recalls that the $\nu = 2/3$ edge profile may involve a more complicated edge reconstruction, as was shown theoretically[6,7] and experimentally[8-10].

A crucial step toward an explanation of an emergent state characterized by $G_{2T} = 2e^2/3h$ was performed by Kane et al.[11,12] (KFP), who allowed random tunneling between the counter-propagating edge modes (due to disorder) accompanied by inter-mode interaction (Fig. 1c, d). A recent theoretical work[13,14] (PGM) expanded the KFP analysis and predicted that, for temperature $T > 0$ and with increasing system length (or, alternatively, with increasing random tunneling strength), the system undergoes a crossover from a clean, non-equilibrated state with two counterpropagating charge modes and $G_{2T} = 4e^2/3h$, to an equilibrated regime with $G_{2T} = 2e^2/3h$ accompanied by neutral modes.

While the existence of a neutral mode, which can transport energy upstream, had been confirmed by Bid et al.[15] and other works[16-19], the full clean-to-equilibrated transition, as predicted by the theory of KFP and PGM, has never been observed. A controlled experimental study of this transition and of the physics involved is missing entirely.

Here, we aim to observe this transition. Our platform is based on a carefully designed double-quantum-well structure (DQW, in a GaAs-based heterostructure), with two populated electronic sub-bands—each tuned separately to the QHE regime (Supplementary Note 1). By top-gating different areas of the structure, the desired counterpropagating modes between $\nu = 1$ and $\nu = 1/3$ can be formed, with a highly controlled inter-mode coupling. We observes the expected full transition of $G_{2T}$ from $4e^2/3h$ to $2e^2/3h$ accompanied by (diffusive) neutral modes.

## Results

**Forming counterpropagating edge modes.** We characterize the system by a generalized filling factor $\nu = (\nu_l, \nu_u)$, where $\nu_l$ ($\nu_u$) is the filling factor in the lower subband, SB1 (higher subband, SB2)[20]. With a clever design of the QW, the densities of the two SBs are misplaced from each other in the growth direction (here, a 0.7-nm-thick AlAs barrier in the center of the QW decreases the coupling strength between the two SBs' wavefunctions). Our device is formed by three horizontal top gates, separating the 2D plane into three regions: upper, center, and lower. Each gate controls the filling factor in the 2DEG underneath it, as shown in Fig. 2a. A 2D plot of the longitudinal resistance, $R_{xx}$, of the upper region (measured when the adjacent region is pinched off), is plotted as function of the magnetic field $B$ and its top-gate voltage, $V_{g1}$ (Fig. 2b, c). The generalized filling factors that correspond to the Hall plateaus are determined by the dark blue regions, where $R_{xx} = 0$, with the current carried by edge modes. Figure 2c is a zoom-in on the "interesting" region, where the generalized filling factors (4/3,0) and (1,1) can be reached by tuning the gates' voltage at a constant magnetic field along the broken yellow lines. For example, at $B = 6$ T, the upper region is at (1,1) at a gate voltage span $V_{g1} = 0.02$–$0.1$ V, and the center region is at (4/3,0) at $V_{g2} = -0.18$ to $\sim -0.2$ V.

By setting the upper and center regions to $\nu = (1,1)$ and $\nu = (4/3,0)$, respectively, the scenario shown in Fig. 1e occurs. The lowest LL of SB1 (e.g., $(1,\uparrow)_{SB1}$), is full in both regions, and thus a $\nu = 1$ edge mode, with spin $\uparrow$, flows along the circumference of the whole region of the sample (with no difficulty to enter the $\nu = 4/3$ state in the center region). The lowest LL of SB2 (e.g., $(1,\uparrow)_{SB2}$), is also full in the upper region and empty in the center region, and thus $\nu = 1$ edge mode with spin $\uparrow$ is flowing only around the upper region, and in the interface between the two regions. Similarly, the second LL of SB1 (e.g., $(1,\downarrow)_{SB1}$) is in $\nu = 1/3$ filling in the center region and empty in the upper region; hence, a $\nu = 1/3$ edge mode with spin $\downarrow$ is flowing only around the center region and counterpropagating at the interface between the regions.

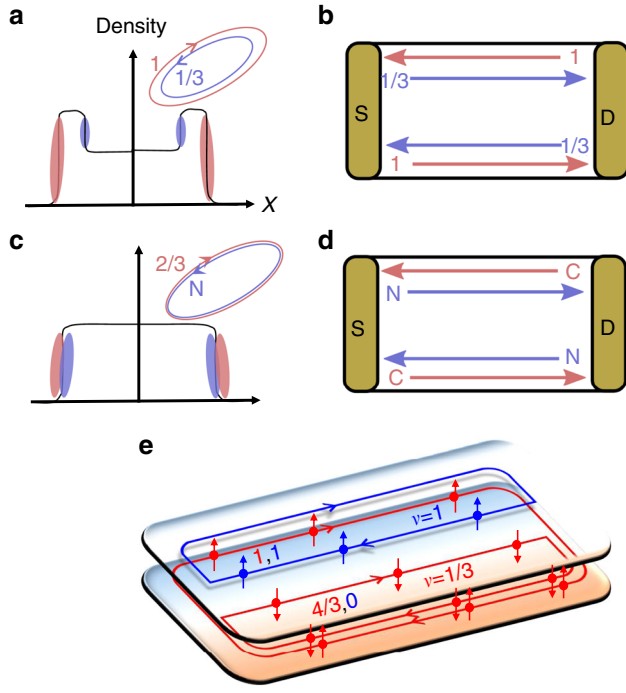

**Fig. 1** The 2/3 hole-conjugate state and its synthetized form. **a** The unequilibrated $\nu = 2/3$ state; composed of two counter-propagating chiral modes: a downstream $\nu = 1$ mode and an upstream $\nu = 1/3$ Laughlin excitation mode. Red and blue represent electron density profile of independent $\nu = 1$ and $\nu = 1/3$ modes at the edges, respectively. **b** Equivalent two-terminal conductance in the unequilibrated regime. **c** The inter-edge scattering results in edge density profile reconstruction: coexistence of a downstream $\nu = 2/3$ mode and an upstream neutral mode. **d** Equivalent two-terminal conductance in the equilibrated regime. **e** Schematics of the device. A $\nu = 1$ edge mode of the first LL belonging to SB1 and having spin $\uparrow$, flows around the whole region of the sample. A $\nu = 1$ edge mode of the first LL belonging to SB2, having spin $\uparrow$, flows only around the upper region. A $\nu = 1/3$ mode of the second LL of SB1, having spin $\downarrow$, flows around the lower region. Thus, at the interface of the two regions, a $\nu = 1$ and a $\nu = 1/3$ modes counterpropagate

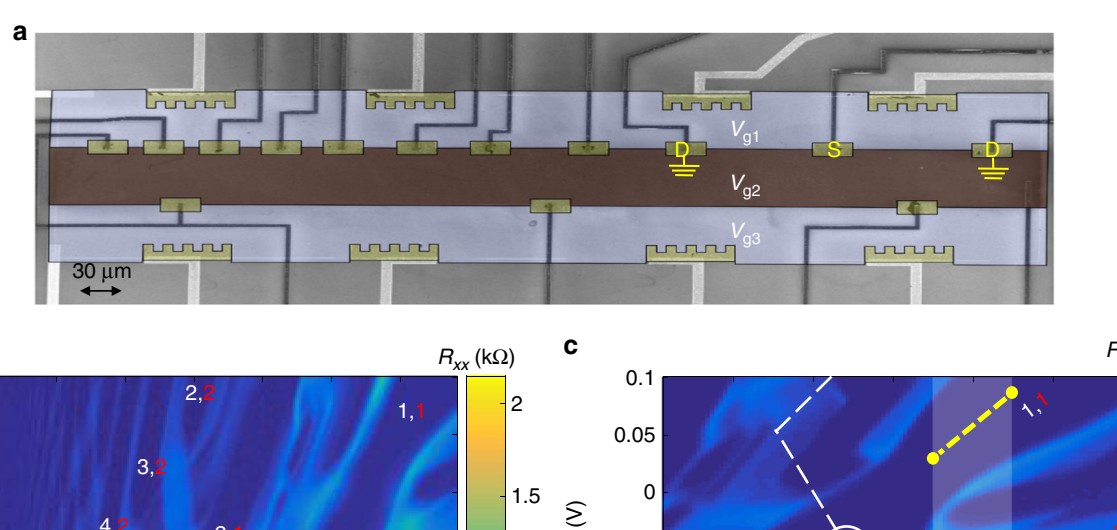

**Fig. 2** Device SEM image and fan diagram for the 2DEG. **a** Scanning electron microscope (SEM) micrograph of the studied device, which is fabricated in a 2DEG embedded in a GaAs/AlGaAs double-quantum well structure. It consists of the upper (light purple), center (brown), and lower (light purple) regions, and the density in each region can be independently controlled by its own top gate, with voltages $V_{g1}$, $V_{g2}$, and $V_{g3}$. A few ohmic contacts are located at the interfaces between two regions (dark yellow squares) and others located away to measure the generalized fillings of the three regions (dark serrated yellow squares). Each source (S) is placed in-between two drains (D) in order to measure two-terminal conductance. **b** 2D mapping of the longitudinal resistance, $R_{XX}$, versus magnetic field and gate voltage $V_{g1}$ at $T = 20$ mK. Data are obtained from the ohmic contacts along the edge of the upper region, in a quantum well with 0.7 -nm-thick AlAs barrier, with the adjacent region pinched off. **c** The zoom-in on the "interesting" region used in our work. The white and red dashed lines represent the non-interacting spin-split Landau levels—LL2 in SB1 and LL1 in SB2, respectively. The field $B_c$ describes the magnetic field where non-interacting subbands cross and hybridize. The clear vertical square (containing the yellow dashed lines) illustrates the region in magnetic field where a transition between the generalized fillings of $v = (1,1)$ and $v = (4/3,0)$ can be tuned

The fan diagram shows LLs belonging to SB1 and SB2, and their revolution as they cross and hybridize in a certain range of magnetic field and gate voltage. The size of the gap is determined by the coupling between the LLs (depends on the lateral separation of the modes and the thickness of the AlAs barrier). Note the emergent gap appearing between the LLs that corresponds to filings $v_l = v_u = 1$ of the (1,1) and (2,0) fillings in Fig. 2c (white circle). One can also observe LLs crossing without an opening of a gap in a similar structure where the AlAs barrier is thicker (Supplementary Note 2).

**Length and magnetic field dependence**. The fabricated device contained a series of ohmic contacts, placed on the interfaces between the top and bottom regions and the center region, being separated by various distances. Each source contact was placed midway between two grounded drains (Fig. 2a). The two-terminal conductance, $G_{2T}$, was measured between source and ground for several source–drains separation lengths, using a standard lock-in technique at fridge temperature of 20 mK. The evolved conductance with the filling in the center region tuned from $v = (1,0)$ to $v = (4/3,0)$, while keeping the upper and lower regions are at $v = (1,1)$, is shown in Fig. 3a for $B = 6.45$ T. It mimics the transition shown in Fig. 1. The highlighted region in red corresponds

to the center region being at filling (1,0), thus supporting a single downstream integer mode (at the top or bottom interfaces) with the conductance equals $e^2/h$—independent of the propagation length. Once the gate voltage of the center region was increased, placing its filling at $v = (4/3,0)$, an evolution of the two-terminal conductance, from $G_{2T} = 4e^2/3h$ at a short distance (6 μm) to $G_{2T} = 2e^2/3h$ at a long distance (150 μm) was observed (highlighted in blue in Fig. 3a).

Tuning the intermode coupling along the yellow dashed lines in Fig. 2c strongly affected the equilibration length. In Fig. 3b, the length dependence of $G_{2T}$ for several magnetic fields is plotted. While at $B = 6.45$ T equilibration sets in around a propagation length of 40 μm, at $B = 5.8$ T it sets in around 6 μm. Figure 3c shows the conductance for a fixed propagation length of 15 μm as a function of the center gate voltage, moving from (1,0) (red region) to (4/3,0) (blue region) for various magnetic fields. In the red region, the conductance is quantized at $e^2/h$, as in Fig. 3a; however, in the blue region, the strong magnetic field dependence is observed. In the high-field region, the conductance approached $G_{2T} = 4e^2/3h$, while at lower field the conductance got close to $G_{2T} = 2e^2/3h$. The observed tilted colored regions illustrate the required gate voltage changes as the magnetic field in order to keep the filling factor constant. Similar behavior, but as a function of field, is illustrated in Fig. 3d.

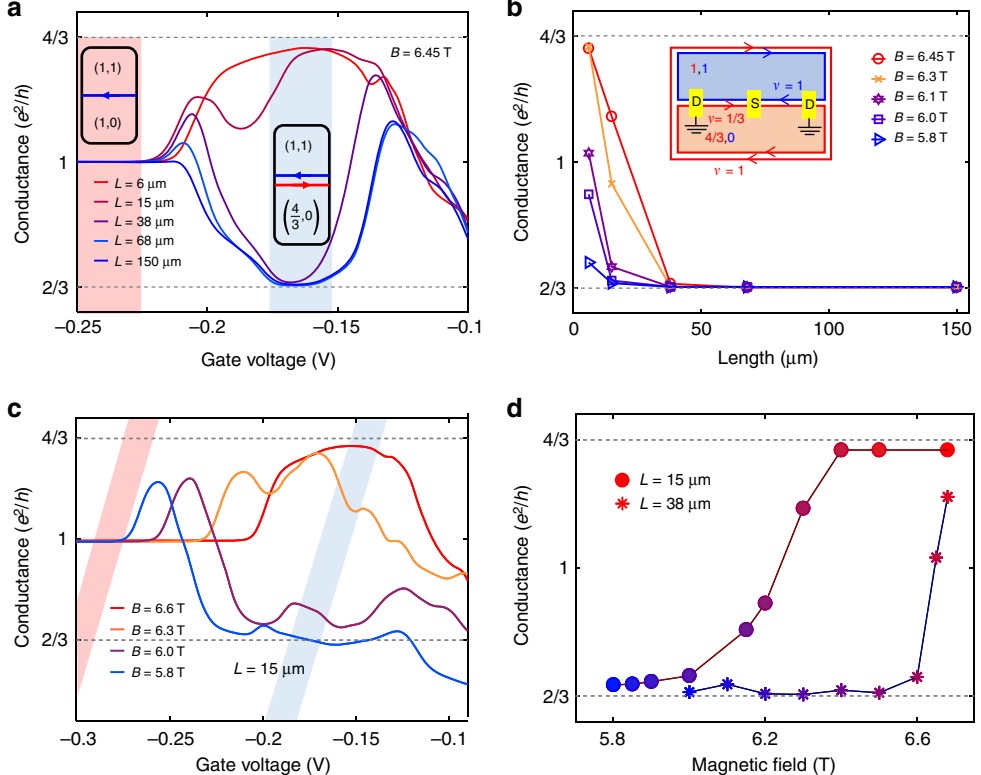

**Fig. 3** Length and magnetic field dependent two-terminal conductance. **a** Two-terminal conductance versus center top-gate voltage for different propagating lengths at $B = 6.45$ T. The upper region is set to (1,1), and the center region is tuned in the ranges of gate voltage highlighted by red and blue areas, tuned to (1,0) and (4/3,0), respectively. In the blue area, a $v = 1$ and a $v = 1/3$ counterpropagating chiral modes coexist at the interface, and the conductance decreases from $G_{2T} = 4e^2/3h$ to $2e^2/3h$ as the channel length increases from 6 μm to 150 μm. In red area, the conductance is $e^2/h$ and is length independent due to a single a $v = 1$ chiral mode at the interface. **b** Two-terminal conductance versus propagating length at different magnetic fields, with the center region is tuned to (4/3,0). **c** Two-terminal conductance of a 15 -μm long channel as a function of center gate voltage in a range in magnetic field 5.8 T < $B$ < 6.6 T. The colored areas are as in panel **c**. **d** The dependence of the two-terminal conductance on the magnetic field for propagating length $L = 38$ μm and 15 μm. With decreasing the magnetic field, the two-terminal conductance evolves from $G_{2T} = 4e^2/3h$ to $2e^2/3h$

These results are consistent with the theoretical prediction[13,14] of a generic (i.e., for any interaction, tunneling, and temperature) crossover in the conductance from $4e^2/3h$ to $2e^2/3h$ with increasing length; the approach to $2e^2/3h$ takes place with exponential accuracy. Further work is needed to accurately study the temperature dependence of the equilibration length, allowing further comparison with theoretical predictions[13]. We also note that the temperature profile along the edge may exhibit interesting structure[14]. These results indicate that the magnetic field may serve as a powerful tool for controlling the intermode equilibration length. We attribute the strong effect of the magnetic field on the equilibration to two main mechanisms: (i) experimentally, it is observed that as both the gate voltage (in the upper region) and the magnetic field lower, a gap emerges (near $B_c$), indicating a stronger equilibration between $v = 1$ mode in SB2 and $v = 1/3$ mode in SB1. Also note that lowering the field increases the magnetic length, thus increasing the overlapping of the wavefunctions; (ii) As the crossing of the two energy dispersions (of the two edge modes) approaches the Fermi energy, the overlapping of the modes in the lateral direction increases (additional bias-dependent conductance measurements are provided in Supplementary Note 3, magnetic field-dependent coupling in the lateral direction in Supplementary Note 5).

**Neutral mode**. According to the theory, the equilibrated quantum Hall state of $v = 2/3$ consists of a downstream charge mode accompanied by a diffusive neutral mode[13,21]. The diffusive propagation of heat at $v = 2/3$ was supported by a recent experiment[22]. Does this neutral mode appear in our synthetic realization of the $v = 2/3$ state? The experimental setup needed to detect the neutral mode by noise measurement is sketched in Fig. 4a. One hot spot located at the back of the source contact, where the voltage drops from V to 0, releases energy that (some of it) propagates upstream via the so-called neutral mode. The injected source current propagated toward the ground, while the voltage noise was measured 38 μm away from the source in A1 and in A2—being indicative of the presence of a heat-carrying neutral mode. Measurements were performed in the equilibrated regime; namely, with $G_{2T} = 2e^2/3h$ and thus charge propagating only downstream (toward A2).

The noise was measured at contact A1 for different magnetic field strengths (see Fig. 4b). The noise increased monotonically with the injected DC current, and tended to saturate at higher current values. As the magnetic field increased (away from $B_c$), the inter-mode interaction got weaker (see Fig. 3), and the measured excess noise increased (see Fig. 4b). The observed noise is a manifestation of the upstream diffusive neutral mode being excited by the hot spot at the back of the source contact (see Fig. 4a). With the magnetic field increasing, the equilibration length, needed to fully excite the neutral mode, increases too, thus facilitating a shorter distance for the heat to reach the amplifier at A1. The more heat arrives the vicinity of A1, the stronger is the intrinsic noise due to the stochastic nature of the backscattering

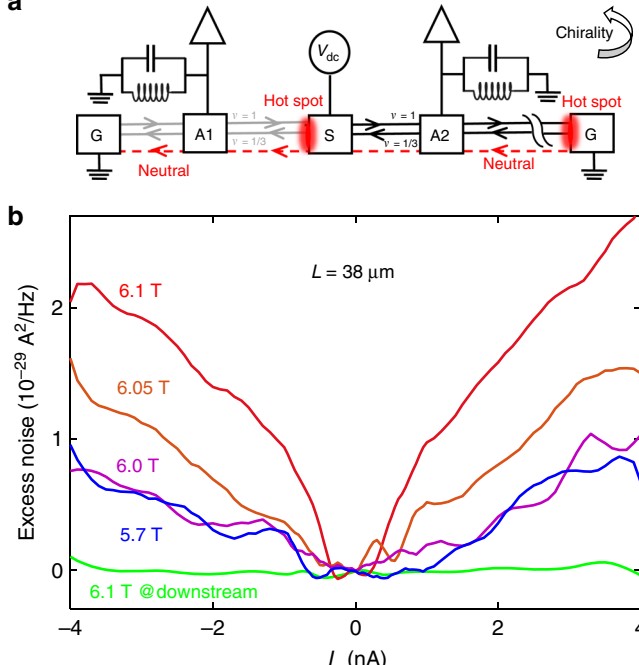

**Fig. 4** Noise measurement setup and experimental results. **a** Schematic diagram of noise measurement circuit. Current is injected from source S and the upstream (downstream) noise is measured by a spectrum analyzer through amplifier contact A1 (A2). The contact is connected to an LC circuit at a center frequency $f_0 = 1.3$ MHz, with the signal amplified by a homemade (cooled) voltage pre-amplifier followed by a commercial, room temperature, voltage amplifier (NF-220F5). Note the gain of cold amplifier is taken as 7.5, but the precise gain value was not precisely calibrated. **b** Upstream excess noise in contact A1 as a function of $I_s$ at different magnetic fields. Green line represents the $I_s$-independent (negligible) downstream excess noise at $B = 6.1$ T

between the edge modes, resulting in a stronger noise signal measured at A1[23]. No sizeable excess noise was detected at A2, which is attributed to the much larger distance between the hot spot at contact G (on the right) and A2 (being 108 μm). As expected, measurements performed when the interface was between (1,1) and (1,0) did not find any upstream excess noise in A1 (Supplementary Note 4).

We successfully fabricated an interface between two counter-propagating chiral modes of filling $\nu = 1$ and $\nu = 1/3$, and controlled their interaction by varying the magnetic field and the electron density. We observed a transition between a two-terminal conductance of $G_{2T} = 4e^2/3\,h$, when intermode interaction was suppressed, and $G_{2T} = 2e^2/3\,h$, when the interaction was strong. We also observed the emergence of an upstream diffusive neutral mode when the conductance approached $G_{2T} = 2e^2/3\,h$, as always observed in the emergent (equilibrated) $\nu = 2/3$ state. These man-made synthetized modes provide a method to study a variety of non-equilibrated FQHE states as well as the transition between their non-equilibrated and equilibrated states.

## Methods

**Sample fabrication**. An etch-defined Hall-bar with Ni/Ge/Au ohmic contacts was fabricated using E-beam lithography. This was followed by an atomic layer deposition of $HfO_2$ followed by an E-gun evaporation of 5/20 nm Ti/Au top gates. The top gates, each defined a part of the 2D plane, were separated by a gap of 80 nm. Finally, the $HfO_2$ is etched in small regions for the ohmic contacts, which were connected to the bonding pads by 5/120 nm Ti/Au leads.

## Data availability

The data that support the plots within this paper and other findings of this study are available from the corresponding author upon reasonable request. The source data for the plots within the article and Supplementary have been deposited in the Harvard Dataverse repository at https://doi.org/10.7910/DVN/SKLH8Y.

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

## Acknowledgements

We acknowledge Erez Berg, Yuval Oreg, Ady Stern, Dmitri Feldman, and Kyrylo Snizhko for fruitful discussions. We thank Diana Mahalu for her help in the E-beam processing, and Vitaly Hanin for his help in the ALD process. M.H. acknowledges the partial support of the Israeli Science Foundation (ISF)—no. 450/16, the Minerva foundation—712598, and the European Research Council under the European Community's Seventh Framework Program (FP7/2007-2013)/ERC, No. 339070. Y.G acknowledges support by DFG grant No. MI 658/10-1, DFG grant No. RO 2247/81, CRC 183 of the DFG, ISF

Grant No. 1349/14, Leverhulme Trust VP-2015-0005 and Italia–Israel project QUAN-TRA. J.P. acknowledges the Koshland Foundation support.

## Author contributions

Y.C., Y.R., and W.Y. contributed equally to this work in a heterostructure design, sample design, device fabrication, measurement setup, data acquisition, data analysis and interpretation, and writing of the paper. D.B. contributed to this work in a heterostructure design and sample design. M.H. contributed in a heterostructure design, sample design, data interpretation, and writing of the paper. J.P., A.D.M., and Y.G. contributed in the data interpretation and writing of the paper. V.U. contributed in a heterostructure design and molecular beam epitaxy growth.

## Additional information

**Competing interests:** The authors declare no competing interests.

