## [Peer Review File · Nature Communications]

Reviewers' comments:

Reviewer #1 (Remarks to the Author):

This manuscript reports on the operation of a device in which lithographic gates are used to bring counter-propagating $\nu=1$ and $\nu=1/3$ edge modes into close proximity to test how these states equilibrate as a function of propagation length and magnetic field. The authors measure the two-terminal

conductance and monitor the system's evolution from $4/3e^2/h$ to $2/3e^2/h$ – from minimal equilibration to full equilibration. The authors' principal conclusions are that the edge modes remain un-equilibrated at short distances but gradually equilibrate over longer length scales due to coupling between the modes. The length scale for equilibration can be modified by tuning the magnetic field and gate voltage. Finally, the authors claim that the equilibrated state with two-terminal

conductance $2/3e^2/h$ is accompanied by the appearance of an upstream neutral mode. The topic of edge state equilibration and edge state reconstructions in the fractional quantum Hall regime is an interesting and timely topic, especially at $\nu=2/3$. The device geometry employed in the current work is also noteworthy and provides a useful tool. Overall, the experiments described in the manuscript will be of interest to researchers working in the field. However, I cannot yet recommend the manuscript for publication until several significant deficiencies are addressed. Some of my concerns are technical for which the authors may possibly provide straightforward explanations. My more serious criticism of this manuscript centers on an overall lack of clarity in description of important experimental details and physics. The reader is often left to guess what the authors mean or even how the experiments were actually conducted. This makes evaluation of the actual merits of work more difficult. The authors should strive to make a clearer and self-contained presentation of the experimental details; this will make for a stronger paper.

Specific questions to be addressed include:

1) The authors give short shrift to description of the heterostructure and its operation that is the basis for the whole experiment. To review this manuscript one must find all details in a previous publication by the same group (reference 20 in the main text). I understand if the authors want to relegate a short discussion to the supplementary information section, but the description provided in the current version of the manuscript is not sufficient to stand as a self-contained presentation of the new experiments. In the supplemental information the authors only show a schematic cartoon of the heterostructure with no real details, and only point the reader to reference 18 of the main text for further details.

Reference 18 turns out to be completely irrelevant to any discussion of the heterostructure used in the current manuscript. This must be a typo; reference 20 is the actual required reading. The details of the coupling between the gate and the individual subbands are important for device operation and interpretation of data of the current manuscript. Is it possible for the authors to give a concise summary of the response to applied gate voltage of the population of the individual subbands at $B=0T$ and at $B=6T$? This would be helpful to understand how the filling factor in each well was determined. Fig. 1b can then be understood more straightforwardly.

2) What do V and O represent in Fig. 1b and Fig. 1d.? These letters sit on the source and drain contacts but are not mentioned in the figure caption or the main text.

3) Based on the text, I inferred that the synthetic $2/3$ edge state is formed by proximity of the $\nu=1$ state in the upper subband (in the upper gated region) and the opposite spin $\nu=1/3$ mode in the lower subband (in the lower gated region). However, this is not explicitly stated in the text. Is this picture correct? It would be beneficial to readers' understanding to explicitly state which spin and which subband each constituent edge mode belongs to.

4) Fig. 3d appears to show that the edge mode is less equilibrated over 38microns than 15microns, which is the opposite the expected trend. Is this correct or is there an error in the legend?

5) A central finding of this work is that the equilibration length between the constituent $\nu=1$ and $\nu=1/3$ modes increases with decreasing magnetic field; however, the authors' proposed explanations need clarification. The authors mention "increased overlap in the vertical direction" as well as "increasing the overlapping of the modes in the lateral direction" at lower field. Can the authors clarify exactly what is meant by these two statements, and why these effects would be expected to occur when the magnetic field is decreased?

6) The section discussing noise caused by a possible upstream neutral mode needs clarification. It is not clear why a neutral mode should cause electronic noise in the

current measurement setup. The authors should make explicit the physical mechanism by which noise is purported to occur. This effect appears to be different than the signatures of neutral modes reported in refs. 15 and 16, where the measured noise was attributed to fragmentation of the neutral mode by a quantum point contact. In the present case there is no QPC, yet the noise is still attributed to a neutral mode. Why isn't a QPC needed in the present case to convert the neutral mode into charge that is registered as noise?

7) The circuit diagram in Fig. 4a is confusing. The authors state that the current injected at the source should flow from source to ground contact, from S to G. However, since the injected current is DC, one would expect all the current to be shunted to ground through the inductors in the resonant tank circuits connected to A1 and A2. The authors should explain the circuit in more detail. Why isn't a blocking capacitor in front of the tank circuit needed? Is there but not shown in the diagram? Perhaps there is some stray circuit capacitance that is utilized?

Reviewer #2 (Remarks to the Author):

Results reported in this manuscript are closely related to the fascinating topic regarding the nature of edge mode at $2/3$ filling fraction. It was first proposed in Ref [4] that edge reconstruction at $2/3$ filling would lead to counter-propagating edge with two terminal conductance of $G=4/3(e^2/h)$. However, two terminal conductance of $G=2/3(e^2/h)$ was observed experimentally, consistent with one single charge mode instead of two. This motivated theoretical prediction of the neutral mode, Ref [11] and [12], which was experimentally confirmed in Ref [15]. In the present manuscript, the authors tried to demonstrate the evolution from counter-propagating charge modes as proposed by Ref [4], to fully equilibrated down-stream charge mode and up-stream neutral mode as previously reported in Ref [15]. Such evolution has been discussed theoretically by Ref [11] [12] and [13], but experimental study is missing.

Accurate control of the position of chiral edge mode is experimentally challenging due to limitations in nano-fabrication techniques, which contributed to the difficulty in the realization and studying of counter-propagating chiral charge states. The authors worked around this problem using a double-quantum-well with intrinsic density imbalance, where electrons are allowed to tunnel between the two quantum wells. In such system it is possible to place counter propagating $\nu=1$ and $\nu=1/3$ edge modes in near vicinity using a split gate geometry, and the coupling strength between the two modes could be controlled by varying channel length and magnetic field.

In order to verify the nature of edge mode, the authors performed electrical transport and noise measurement. By tuning the coupling strength between the two counter-propagating edges, a transition is demonstrated between non-equilibrated modes with 2-terminal conductance of $G=4/3(e^2/h)$ to fully equilibrated modes with conductance of $G=2/3(e^2/h)$. Additionally, in the fully equilibrated limit, an up-stream neutral mode is observed in the form of excess noise, which is consistent with prior measurements in Ref [17].

Overall, the authors utilized a novel design to address an outstanding question related to fractional quantum hall physics, and the data presented in this work is convincing. The manuscript deserves publication in Nature communication, if the authors could address the following comments.

(1) A double quantum well or wide single quantum well is fundamentally different from an ideal single 2D confinement, which are studied extensively in the context of multi-component quantum Hall states. It has been demonstrated that the nature of ground states in a double quantum well is sensitive to parameters such as interlayer separation, density imbalance and tunneling strength (Δ_{SAS}) (e.g. this is studied in PHYSICAL REVIEW B 88, 245413 (2013)). A thorough characterization of the double quantum well is critical to understanding the coupling between the counter-propagating edge modes, and should be included in the manuscript.

(2) A similar double quantum well is reported in Ref. [20], where the ground states are identified based on the fan diagram. This identification is subsequently confirmed by two-terminal measurement along different edge modes. However, the double quantum well in this work is

significantly different, with a barrier that is 0.7 nm thick, compared to 3nm in Ref [20]. A thinner barrier is going to change the charge carrier distribution between the two quantum, resulting in increased wavefunction overlap and tunneling between the two subbands. The effect of tunneling is evidenced by the difference between the fan diagram in Fig.2b and Ref. [20]. Some discussion should be included regarding how ground states are identified here, and how this identification is influenced by the thinner barrier and increased tunneling. A specific suggestion is to show the full fan diagram in a larger gate and magnetic field range, as Fig.2e of Ref. [20]. Additionally Hall resistance or two-terminal resistance measured along the $1/3$ (or $1+1/3$) edge mode in the absence of a counter-propagating mode will be valuable information as well.

(3) An emergent gap is observed in the white circle of Fig.2b, which is attributed to increased tunneling. It is not immediately clear why increased tunneling will lead to an emergent energy gap and what the nature of this gap is. Since this gap is a prominent feature resulting from the specific design of the double quantum well (thin barrier), I think it is fair to request some discussion about this emergent gap to be included in the manuscript.

(4) Changing barrier thickness should have a big impact on the coupling between counter-propagating edge modes. Is this something that the authors looked into? Judging from Fig.2e of Ref [20], counter-propagating chiral edge could be realized in the device with 3nm thick barrier, and it will be very interesting to find out how edge equilibration changes with a 3nm barrier.

(5) Fig.3a is measured at constant magnetic field of $B=6.45T$. However, the $4/3$ state is not fully developed at $B=6.45T$ according to Fig.2b. Is this because the fan diagram in the center region is shifted compared to the upper region? What does this fan diagram look?

(6) Fig.4 plots noise measurement in the full-equilibration limit. The comparison between excess noise measured up-stream and down-stream is very convincing, but I'm very curious about what noise measurements look like in the limit where $\nu=1$ and $\nu=1/3$ mode do not equilibrate. It should be included here if such measurements exist.

A few minor comments:

(1) On the second line of summary, it should be "electron density" instead of "elector density".

(2) The authors mentioned that the 0.7nm barrier is intended to decrease density coupling between the two subbands. I assume that density coupling is means decreases tunneling amplitude. This should be clarified as density coupling is somewhat ambiguous.

(3) In supplementary information section SI, I don't think Ref [18] is the intended reference.

(4) In supplementary information section S3, it states that differential conductance is shown in Fig.S3, but the figure axis is labeled conductance.

Reviewer #3 (Remarks to the Author):

The authors construct a "synthetic" $\nu = 2/3$ edge by building a device that effectively puts two distinct quantum Hall states in contact with each other. The first state has two layers each at $\nu = 1$ and the second state has one layer at $\nu = 0$ and the other at $\nu = 4/3$. The interface between the two regions consists of counter propagating $\nu = 1$ and $\nu = 1/3$ edges, thus forming an effective $\nu = 2/3$ edge. This is a creative way to build and manipulate a $\nu = 2/3$ edge.

The authors then show that they can tune between the different edge phase (put forth by Kane, Fisher, and Polchinski) by measuring the Hall conductance at the interface. This is a novel result because earlier results did not tune between these phases: usually, the physics of the edge is fixed by the experimental details that are difficult to manipulate.

However, in some instance the authors have stated their results and methods without much explanation. Furthermore, a comparison to theoretical results (especially Refs 13 and 14, which study the crossover at $\nu = 2/3$) is lacking.

Therefore, I recommend the paper for publication in Nature Communications after the following points have been addressed:

1. Was it essential to combine $\nu = 2$ and $\nu = 4/3$ instead of $\nu = 1$ and $\nu = 1/3$? Why?
2. What is the role of the third region (lower) region? The authors mention that the lower region is also tuned to two $\nu = 1$ layers and then never discuss it again.
3. In the middle of p5, the authors write, "Tuning the inter-mode coupling along the yellow dashed lines..." However, it is not the inter-mode coupling that is being tuned, but rather B and V . The authors should be more clear here about what knob is being tuned in the experiment vs what effective term in the Hamiltonian they expect is changing.
4. On the top of p6, the authors discuss two mechanisms for how changing the magnetic field might affect the wave function overlap (both vertically and laterally.) A more detailed discussion would benefit this section: in both cases, why does changing B affect the wave function overlap?
5. The noise measurement indicates the presence of counterpropagating edge modes, but the discussion also raises several questions: is there a reason why the noise saturates? Why does it change with B and why would it change with interactions? Is it possible to compare the results with theory? There are several theoretical papers discussing noise in the $\nu = 2/3$ state, none of which are cited, for example:
 - Takei and Rosenow Phys. Rev. B 84, 235316 (2011)
 - Takei, Rosenau, Stern Phys. Rev. B 91, 241104(R) (2015)
 - Shtanko, Snizhko, Cheianov Phys. Rev. B 89, 125104 (2014)
 - Cano and Nayak Phys. Rev. B 90, 235109 (2014)
6. Fig 2b is labelled by V ; does V mean $V_{\{g1\}}$, $V_{\{g2\}}$ or $V_{\{g3\}}$? The text says $V_{\{g1\}}$, but the caption says that $V_{\{g1\}}$ tunes only the upper region. Can the authors clarify which gate voltage is being tuned and how the filling fractions change or remain constant in each region when tuning the gate?
7. In Figs 3a, 3b and 4b, is V or I being continuously tuned? If not, can the authors show discrete data points?
8. There is a typo in the caption to figure 3a, where it reads, " $G_{\{2T\}} = (4/3)e^2/h$ to $(2/3)e^2/3$ ". The last term should be $(2/3)e^2/h$.
9. After mentioning that Refs 13 and 14 predict a crossover between the different $\nu = 2/3$ states, the results of those references are never mentioned again. Is it possible to more specifically state the results of those references and compare them to the experimental results?

The reply to the Reviewers' comments are presented below. Our response is in blue.

The main changes and additions in the main text are in red – which in most cases are copied within the Reply to the Reviewers.

Reviewer #1:

This manuscript reports on the operation of a device in which lithographic gates are used to bring counter-propagating $\nu=1$ and $\nu=1/3$ edge modes into close proximity to test how these states equilibrate as a function of propagation length and magnetic field. The authors measure the two-terminal conductance and monitor the system's evolution from $4/3e^2/h$ to $2/3e^2/h$ – from minimal equilibration to full equilibration. The authors' principal conclusions are that the edge modes remain un-equilibrated at short distances but gradually equilibrate over longer length scales due to coupling between the modes. The length scale for equilibration can be modified by tuning the magnetic field and gate voltage. Finally, the authors claim that the equilibrated state with two-terminal conductance $2/3e^2/h$ is accompanied by the appearance of an upstream neutral mode.

The topic of edge state equilibration and edge state reconstructions in the fractional quantum Hall regime is an interesting and timely topic, especially at $\nu=2/3$. The device geometry employed in the current work is also noteworthy and provides a useful tool. Overall, the experiments described in the manuscript will be of interest to researchers working in the field. However, I cannot yet recommend the manuscript for publication until several significant deficiencies are addressed. Some of my concerns are technical for which the authors may possibly provide straightforward explanations. My more serious criticism of this manuscript centers on an overall lack of clarity in description of important experimental details and physics. The reader is often left to guess what the authors mean or even how the experiments were actually conducted. This makes evaluation of the actual merits of work more difficult. The authors should strive to make a clearer and self-contained presentation of the experimental details; this will make for a stronger paper.

Specific questions to be addressed include:

- 1) The authors give short shrift to description of the heterostructure and its operation that is the basis for the whole experiment. To review this manuscript one must find all details in a previous publication by the same group (reference 20 in the main text). I understand if the authors want to relegate a short discussion to the supplementary information section, but the description provided in the current version of the manuscript is not sufficient to stand as a self-contained presentation of the new experiments. In the supplemental information the authors only show a schematic cartoon of the heterostructure with no real details, and only point the reader to reference 18 of the main text for further details.

Reference 18 turns out to be completely irrelevant to any discussion of the heterostructure used in the current manuscript. This must a typo; reference 20 is the actual

required reading. The details of the coupling between the gate and the individual subbands are important for device operation and interpretation of data of the current manuscript. Is it possible for the authors to give a concise summary of the response to applied gate voltage of the population of the individual subbands at $B=0T$ and at $B=6T$? This would be helpful to understand how the filling factor in each well was determined. Fig. 1b can then be understood more straightforwardly.

To make the description more transparent, we added more detailed information regarding the heterostructure and electron population of the individual subbands with applied gate voltage and magnetic field.

1. Indeed reference #20 is the correct paper that we should cite in section S1 in the Supplementary Information.

2. The following sentence was added (in the sixth paragraph in the main text) to illustrate how to tune the gate voltage in order to reach the $\nu=(3/4,0)$ and $\nu=(1,1)$ in the upper and center regions, respectively:

“For example, at $B=6T$, the upper region is at $(1,1)$ at a gate voltage span $V_{g1}=0.02\sim 0.1V$, and the center region is at $(4/3,0)$ at $V_{g2}=-0.18\sim -0.2 V$ ”.

3. Here we include also a reply to the third comment of the Reviewer (see below), by changing the following (6th paragraph in the main text):

“By setting the upper and center regions to $\nu=(1,1)$ and $\nu=(4/3,0)$, respectively, we induced two counter-propagating modes at the interface between the two regions: a $\nu=1$ mode flowing to the left and a $\nu=1/3$ mode flowing to the right (Fig. 1e)”

To the following text:

“By setting the upper and center regions to $\nu=(1,1)$ and $\nu=(4/3,0)$, respectively, the scenario shown in Fig. 1e occurs. The lowest LL of SB1 (*i.e.*, $(1, \uparrow)_{SB1}$), is full in both regions, and thus a $\nu=1$ edge mode, with spin \uparrow , flows along the circumference of the whole region of the sample. The lowest LL of SB2 (*i.e.*, $(1, \uparrow)_{SB2}$), is full in the upper region and empty in the center region, and thus $\nu=1$ edge mode with spin \uparrow is flowing only around the upper region, and in the interface between the two regions. Similarly, the second LL of SB1 (*i.e.*, $(1, \downarrow)_{SB1}$) is in $\nu=1/3$ filling in the center region and empty in the upper region; hence, a $\nu=1/3$ edge mode with spin \downarrow is flowing only around the center region and counter-propagating at the interface between the regions.”

2) What do V and O represent in Fig. 1b and Fig. 1d.? These letters sit on the source and drain contacts but are not mentioned in the figure caption or the main text.

These letters meant to suggest the biasing with voltage V and 0. To make it clear, we change V and 0 to Source (S) and Drain (D).

3) Based on the text, I inferred that the synthetic $2/3$ edge state is formed by proximity of the $\nu=1$ state in the upper subband (in the upper gated region) and the opposite spin $\nu=1/3$ mode in the lower subband (in the lower gated region). However, this is not explicitly stated in the text. Is this picture correct? It would be beneficial to readers' understanding to explicitly state which spin and which subband each constituent edge mode belongs to.

We believe that the proposed change (above) in the main text clarifies this question too. We also modified the corresponding text in the Caption of Fig. 1e.

“Figure 1e. Schematics of the device. A $\nu=1$ edge mode of the first LL belonging to SB1, having spin \uparrow , flows around the whole region of the sample. A $\nu=1$ edge mode of the first LL belonging to SB2, having spin \uparrow , flows around the upper region. A $\nu=1/3$ mode of the second LL of SB1, having spin \downarrow , flows around the center region. Thus, at the interface of the two regions, a $\nu=1$ and a $\nu=1/3$ modes counter propagate.

4) Fig. 3d appears to show that the edge mode is less equilibrated over 38 microns than 15 microns, which is the opposite the expected trend. Is this correct or is there an error in the legend?

There is indeed a mistake in the labeling, which we corrected.

5) A central finding of this work is that the equilibration length between the constituent $\nu=1$ and $\nu=1/3$ modes increases with decreasing magnetic field; however, the authors' proposed explanations need clarification. The authors mention "increased overlap in the vertical direction" as well as "increasing the overlapping of the modes in the lateral direction" at lower field. Can the authors clarify exactly what is meant by these two statements, and why these effects would be expected to occur when the magnetic field is decreased?

This is mainly an experimental fact. Looking at the fan diagram in Fig. 2c, the overlap regions between the two fillings is limited (dotted yellow lines). Hence, we must control the coupling in these regions. Observing the region flowing from $\nu=(1,1)$ to $\nu=(2,0)$, we note the gap widening as the magnetic field and the gate voltage decrease (near B_C) – suggesting stronger coupling (and the $\nu=4/3$ state in SB1 may behave similarly as the $\nu=2$) around closer to B_C . We assume that in the present conditions of tuning the parameters, the 'overlap' of the wavefunctions increase as the states' corresponding energies are closer to degeneracy.

Also, at lower fields the magnetic length gets larger, and thus coupling between the two-side wavefunctions increases.

We provide below a suggestive plot of the edge states' energies in the lateral direction. In our measurement, the Fermi level must decrease with decreasing magnetic fields to keep a constant filling factor. Consequently, as the strongest coupling takes place when the 'crossing point' of the LLs lays at (or near) the Fermi energy, it is likely that by lowering the field and gate voltage brings us close to that optimal condition (the lowest field is restricted, as can be seen from the fan diagram).

We thus modified the main text to explain increased overlap along these lines:

We attribute the strong effect of the magnetic field on the equilibration to two main mechanisms: *i.* Experimentally, it is observed that as both, the gate voltage (in the upper region) and the magnetic field lower, a gap emerges (near B_c) – indicating a stronger equilibration between $\nu=1$ mode in SB2 and $\nu=1/3$ mode in SB1. Note also that lowering the field increases the magnetic length, thus increasing the overlapping of the wavefunctions; *ii.* As the crossing of the two energy dispersions (of the two edge modes) approaches the Fermi energy, the overlapping of the modes in the lateral direction increases (additional bias dependent conductance measurements are provided in the Supplementary Section – S3, magnetic field dependent coupling in the lateral direction in the Supplementary Section – S5).

We also added the section “Magnetic field dependent coupling in the lateral direction” in the Supplementary Information:

Figure S5 is a suggestive plot of the edge states’ energies in the lateral direction. In our measurement, the Fermi level must decrease with decreasing magnetic fields to keep a constant filling factor. Consequently, as the strongest coupling takes place when the ‘crossing point’ of the LLs lays at (or near) the Fermi energy, it is likely that by lowering the field and gate voltage brings us close to that optimal condition (the lowest field is restricted, as can be seen from the fan diagram).

Figure S5. A schematic diagram of the edge states' energies along the y direction. By decreasing magnetic field and gate voltage to make the Fermi level get close to the 'crossing point' of the LLs, the stronger coupling can be achieved.

6) The section discussing noise caused by a possible upstream neutral mode needs clarification. It is not clear why a neutral mode should cause electronic noise in the current measurement setup. The authors should make explicit the physical mechanism by which noise is purported to occur. This effect appears to be different than the signatures of neutral modes reported in refs. 15 and 16, where the measured noise was attributed to fragmentation of the neutral mode by a quantum point contact. In the present case there is no QPC, yet the noise is still attributed to a neutral mode. Why isn't a QPC needed in the present case to convert the neutral mode into charge that is registered as noise?

As pointed out by the Reviewer, the mechanism considered here has a different origin compared with previously discussed noise-generating mechanisms (Refs. 15 and 16). In the latter, fragmentation of neutral mode by a QPC gives rise to noise (with zero DC current). The observed noise here is due to the upstream propagation of a diffusive neutral mode excited by the hot spot (highlighted in Fig. 4a). The heat is diffusively transported towards A1. This heat stimulates stochastic electron tunneling between the counter-propagating (original) modes; hence stochastic backscattering of charge emitted from A1. More details can be found in the theory paper arXiv: 1810.06871. We also modified Fig. 4a to clarify the source of the upstream noise.

We have now added the corresponding clarifications in the Section "Neutral Mode". We have also added the above arXiv paper to the reference list.

We assume that the thermal noise caused by heating the amplifier contact by the upstream heat flow, is likely to be very small due to the large size of the amplifier's contact. Note: We removed the curve at 6.2T, as it was on the boundary allowed by the fan diagram.

The original text is:

“The experimental setup needed to measure the neutral mode is sketched in Fig. 4a”

We changed it:

The experimental setup needed to detect the neutral mode by noise measurement is sketched in Fig. 4a. One hot-spot located at the back of the Source contact, where the voltage drops from V to 0, releases energy that (some of it) propagates upstream via the so called neutral mode.

7) The circuit diagram in Fig. 4a is confusing. The authors state that the current injected at the source should flow from source to ground contact, from S to G. However, since the injected current is DC, one would expect all the current to be shunted to ground through the inductors in the resonant tank circuits connected to A1 and A2. The authors should explain the circuit in more detail. Why isn't a blocking capacitor in front of the tank circuit needed? Is there but not shown in the diagram? Perhaps there is some stray circuit capacitance that is utilized?

We indeed had a blocking capacitor that prevents shoring it through the coil. Note, though, that the measured noise is independent of the presence of the blocking capacitor.

Reviewer #2:

Results reported in this manuscript are closely related to the fascinating topic regarding the nature of edge mode at $2/3$ filling fraction. It was first proposed in Ref [4] that edge reconstruction at $2/3$ filling would lead to counter-propagating edge with two terminal conductance of $G=4/3(e^2/h)$. However, two terminal conductance of $G=2/3(e^2/h)$ was observed experimentally, consistent with one single charge mode instead of two. This motivated theoretical prediction of the neutral mode, Ref [11] and [12], which was experimentally confirmed in Ref [15]. In the present manuscript, the authors tried to demonstrate the evolution from counter-propagating charge modes as proposed by Ref [4], to fully equilibrated downstream charge mode and up-stream neutral mode as previously reported in Ref [15]. Such evolution has been discussed theoretically by Ref [11] [12] and [13], but experimental study is missing.

Accurate control of the position of chiral edge mode is experimentally challenging due to limitations in nano-fabrication techniques, which contributed to the difficulty in the realization and studying of counter-propagating chiral charge states. The authors worked around this problem using a double-quantum-well with intrinsic density imbalance, where electrons are allowed to tunnel between the two quantum wells. In such system it is possible to place counter propagating $\nu=1$ and $\nu=1/3$ edge modes in near vicinity using a split gate geometry, and the

coupling strength between the two modes could be controlled by varying channel length and magnetic field.

In order to verify the nature of edge mode, the authors performed electrical transport and noise measurement. By tuning the coupling strength between the two counter-propagating edges, a transition is demonstrated between non-equilibrated modes with 2-terminal conductance of $G=4/3(e^2/h)$ to fully equilibrated modes with conductance of $G=2/3(e^2/h)$. Additionally, in the fully equilibrated limit, an up-stream neutral mode is observed in the form of excess noise, which is consistent with prior measurements in Ref [17].

Overall, the authors utilized a novel design to address an outstanding question related to fractional quantum hall physics, and the data presented in this work is convincing. The manuscript deserves publication in Nature communication, if the authors could address the following comments.

(1) A double quantum well or wide single quantum well is fundamentally different from an ideal single 2D confinement, which are studied extensively in the context of multi-component quantum Hall states. It has been demonstrated that the nature of ground states in a double quantum well is sensitive to parameters such as interlayer separation, density imbalance and tunneling strength (Δ_{SAS}) (e.g. this is studied in PHYSICAL REVIEW B 88, 245413 (2013)). A thorough characterization of the double quantum well is critical to understanding the coupling between the counter-propagating edge modes, and should be included in the manuscript.

Our understanding emanates mostly by trial and error, where the ground state of the double quantum wells had been be studied by measuring the fan diagram (the $R_{xx}=0$ regions – in blue - represent quantized conductance, as seen in Figs. 2b & 2c). The exact filling factors were extracted via the Hall resistance in the blue regions. We studied 47 nm wide quantum well with different thickness of the center barrier (3nm, 1.5nm and 0.7nm). We found well-distinguished two subbands in all these quantum wells, even though detailed differences were obviously observed due to the barrier's thickness (related intra-band coupling). For example, an emergent gap appears in a quantum well with 0.7nm barrier width (Figs. 2b & 2c) but disappear in a quantum well with a thicker barrier (Fig. S2b). Moreover, no coupling between counter-propagating 1 and $\frac{1}{3}$ modes was observed in quantum wells with 3nm and 1.5nm thick barriers, while it occurs in 0.7nm thick barrier - described in the main text.

(2) A similar double quantum well is reported in Ref. [20], where the ground states are identified based on the fan diagram. This identification is subsequently confirmed by two-terminal measurement along different edge modes. However, the double quantum well in this work is significantly different, with a barrier that is 0.7 nm thick, compared to 3nm in Ref [20]. A thinner barrier is going to change the charge carrier distribution between the two quantum, resulting in

increased wavefunction overlap and tunneling between the two subbands. The effect of tunneling is evidenced by the difference between the fan diagram in Fig.2b and Ref. [20]. Some discussion should be included regarding how ground states are identified here, and how this identification is influenced by the thinner barrier and increased tunneling. A specific suggestion is to show the full fan diagram in a larger gate and magnetic field range, as Fig.2e of Ref. [20]. Additionally Hall resistance or two-terminal resistance measured along the $1/3$ (or $1+1/3$) edge mode in the absence of a counter-propagating mode will be valuable information as well.

We added a fan diagram in a wide range of parameters, in particular in a range of magnetic field 0-7T in our 47nm wide well with a 0.7nm thick barrier (Fig. 2b). The filling factors are labeled based on measured Hall resistance. We also added the dependence of the Hall resistance in the center-gate voltage V_{g2} when the upper region is at (1,1) in Fig. S6 in the Supplementary Information. The observed Hall resistances are 25.8kOhm and 38.7kOhm, representing the fillings in the central region (1,0) and (4/3,0), respectively.

Fig. 2. b. The longitudinal resistance, R_{xx} , as a function of magnetic field and gate voltage, in our DQW with 0.7nm thick AlAs barrier.

Fig. S6. Hall resistance in a DQW with 0.7 nm thick AlAs barrier, R_{xy} , vary with top-gate gate of the central region at different magnetic fields. In this configuration, the filling factor of upper region is (1,1), thus the observed R_{xy} of 25.8kOhm and 38.7kOhm represent the central region at fillings (1,0) and (4/3,0), respectively.

(3) An emergent gap is observed in the white circle of Fig.2c, which is attributed to increased tunneling. It is not immediately clear why increased tunneling will lead to an emergent energy gap and what the nature of this gap is. Since this gap is a prominent feature resulting from the specific design of the double quantum well (thin barrier), I think it is fair to request some discussion about this emergent gap to be included in the manuscript.

The emergent gap comes from the hybridization of SB1 in the second LL and the SB2 in the first LL. As the magnetic field changes, these LLs evolve with different slopes and, due to mixing, they repel each other (changing thus their nature), with a resulting gap in the bulk. This bulk gap lead to $R_{xx}=0$. The structure of the double quantum well (mainly, on the thickness of the barrier), will determine the size of the gap.

We thus added this sentence in 8th paragraph in the main text:

“The fan diagram shows LLs belonging to SB1 and SB2, as they cross and thus repel each other in a certain range of magnetic field. The size of the gap is determined by the hybridization of the LLs (depending on the thickness of the barrier).”

(4) Changing barrier thickness should have a big impact on the coupling between counter-propagating edge modes. Is this something that the authors looked into? Judging from Fig.2e of

Ref [20], counter-propagating chiral edge could be realized in the device with 3nm thick barrier, and it will be very interesting to find out how edge equilibration changes with a 3nm barrier.

Since the thickness of the barrier has a strong effect on inter-LL coupling, we have checked different structures of a 47nm wide quantum well, with three different barrier thicknesses of 3nm, 1.5nm and 0.7nm. Significant coupling (in reasonable propagation lengths) was found only in the quantum well with a 0.7nm barrier, which we employed in this paper. Coupling in the 3nm barrier case was not be observed - in the 1.5nm barrier case it was observed in very long propagation distances and in a limited span of magnetic field - thus not allowing observation of length dependence and magnetic field dependence with a 3nm barrier.

(5) Fig.3a is measured at constant magnetic field of $B=6.45T$. However, the $4/3$ state is not fully developed at $B=6.45T$ according to Fig.2b. Is this because the fan diagram in the center region is shifted compared to the upper region? What does this fan diagram look?

As can be seen in Fig. 3a, the exact quantization at the lowest gate voltage (with a single integer edge mode) suggests that there is no shift in the filling factors in the different regions. The lack of an exact quantization suggests that, even at the shortest distance some finite equilibration takes place (in all Figs. 3). It seems that in the range of controlling parameters at our disposal, this minute equilibration is unavoidable.

(6) Fig.4 plots noise measurement in the full-equilibration limit. The comparison between excess noise measured up-stream and down-stream is very convincing, but I'm very curious about what noise measurements look like in the limit where $\nu=1$ and $\nu=1/3$ mode do not equilibrate. It should be included here if such measurements exist.

Unfortunately, we did not have this data. We do not expect noise when the $\nu=1$ and $\nu=1/3$ mode are decoupled – since we measure nearly exact quantization of $4e^2/3h$ – and thus no other edge modes are possible. Yet, any minute equilibration (that will slightly deviate the conductance from its full quantized level) may always take place (at this distance), and lead to some residual noise.

A few minor comments:

(1) On the second line of summary, it should be “electron density” instead of “elector density”.

We fixed this line in the summary.

(2) The authors mentioned that the 0.7nm barrier is intended to decrease density coupling between the two subbands. I assume that density coupling is means decreases tunneling amplitude. This should be clarified as density coupling is somewhat ambiguous.

We rewrote it to “decrease the electron tunneling between the two subbands”

(3) In supplementary information section SI, I don't think Ref [18] is the intended reference.

Reference 20 is the correct paper that we should cite. This was fixed in the text.。

(4) In supplementary information section S3, it states that differential conductance is shown in Fig. S3, but the figure axis is labeled conductance.

We corrected this as follows:

Reviewer #3:

The authors construct a "synthetic" $\nu = 2/3$ edge by building a device that effectively puts two distinct quantum Hall states in contact with each other. The first state has two layers each at $\nu = 1$ and the second state has one layer at $\nu = 0$ and the other at $\nu = 4/3$. The interface between the two regions consists of counter propagating $\nu = 1$ and $\nu = 1/3$ edges, thus forming an effective $\nu = 2/3$ edge. This is a creative way to build and manipulate a $\nu = 2/3$ edge.

The authors then show that they can tune between the different edge phase (put forth by Kane, Fisher, and Polchinski) by measuring the Hall conductance at the interface. This is a novel result because earlier results did not tune between these phases: usually, the physics of the edge is fixed by the experimental details that are difficult to manipulate.

However, in some instance the authors have stated their results and methods without much explanation. Furthermore, a comparison to theoretical results (especially Refs 13 and 14, which study the crossover at $\nu = 2/3$) is lacking.

Therefore, I recommend the paper for publication in Nature Communications after the following points have been addressed:

1. Was it essential to combine $\nu = 2$ and $\nu = 4/3$ instead of $\nu = 1$ and $\nu = 1/3$? Why?

In our case we used the transition from $\nu=(1,1)$ to $\nu=(4/3,0)$ where the magnetic field is around 6T. The transition from $\nu=(0,1)$ to $\nu=(1/3,0)$ – as the Reviewer suggested - would have given the desired counter-propagating modes structure and would be certainly very interesting to explore. Yet, it cannot be reached with our devices as it would have required an unrealistic magnetic field and gate voltage.

2. What is the role of the third region (lower) region? The authors mention that the lower region is also tuned to two $\nu = 1$ layers and then never discuss it again.

The lower region, which is indeed set to (1,1) like the upper region, plays the same role as the upper region and simply has contacts with larger separation distance at its interface with the center region, thus allowing one more (1,1) to (4/3,0) interface in the same device.

3. In the middle of p5, the authors write, "Tuning the inter-mode coupling along the yellow dashed lines..." However, it is not the inter-mode coupling that is being tuned, but rather B and V. The authors should be more clear here about what knob is being tuned in the experiment vs what effective term in the Hamiltonian they expect is changing.

We agree. We change the main corresponding text to clarify that point.

The original text:

“Tuning the inter-mode coupling along the yellow dashed lines in Fig. 2c, affected strongly the equilibration length.”

The new text:

“Tuning the magnetic field and gate voltage along the yellow dashed lines in Fig. 2c affected strongly the inter-mode coupling and the equilibration length.”

4. On the top of p6, the authors discuss two mechanisms for how changing the magnetic field might affect the wave function overlap (both vertically and laterally.) A more detailed discussion would benefit this section: in both cases, why does changing B affect the wave function overlap?

This is mostly an experimental observation. We think our response to the 6th comment of Reviewer #2 clarified this question too. Our changes included in the section “Length and magnetic field dependence” in the main text and in S5 in the Supplementary Information.

5. The noise measurement indicates the presence of counter-propagating edge modes, but the discussion also raises several questions: is there a reason why the noise saturates? Why does it change with B and why would it change with interactions? Is it possible to compare the results with theory? There are several theoretical papers discussing noise in the $\nu = 2/3$ state, none of which are cited, for example:

- Takei and Rosenow Phys. Rev. B 84, 235316 (2011)
- Takei, Rosenau, Stern Phys. Rev. B 91, 241104(R) (2015)
- Shtanko, Snizhko, Cheianov Phys. Rev. B 89, 125104 (2014)
- Cano and Nayak Phys. Rev. B 90, 235109 (2014)

A very recent theoretical analysis (cf. arXiv:1810.06871) indicates that the noise scales as $\sqrt{\ell_{\text{eq}}/L}$, where L is the length between the hot-spot and the contact at which the noise is measured, and ℓ_{eq} is the inter-mode equilibration length. This scaling reflects the diffusive nature of heat propagation, along with the ballistic propagation of the charge mode. This theoretical prediction is consistent (at least qualitatively) with the noise measurements. With the magnetic field B increasing, the magnetic length decreases, rendering the overlap between the chiral modes smaller, hence decreasing the inter-mode tunneling. This implies that the equilibration length increases. This increase of the equilibration length with B is clearly seen in the conductance data in Fig. 3. The enhancement of noise with B (i.e. with ℓ_{eq}) as observed in Fig. 4 is thus consistent with theoretical expectations.

The theory papers cited by the Reviewer are very interesting, but they address a different mechanism of noise generation, which involves a setup with a QPC.

We have added the above theoretical arXiv paper to the reference list. We have also added the corresponding clarifications in the Section “Neutral Mode”:

“The more heat arrives the vicinity of A1, the stronger is the intrinsic noise due to the stochastic nature of the backscattering between the edge modes, resulting in a stronger noise signal measured at A1.”

6. Fig 2b is labelled by V ; does V mean $V_{\{g1\}}$, $V_{\{g2\}}$ or $V_{\{g3\}}$? The text says $V_{\{g1\}}$, but the caption says that $V_{\{g1\}}$ tunes only the upper region. Can the authors clarify which gate voltage is being tuned and how the filling fractions change or remain constant in each region when tuning the gate?

The label V means V_{g1} in Fig. 2c. Conductance was measured between contacts in the upper region when the gate voltage V_{g1} was scanned, while the adjacent region was depleted. Thus the fan diagram represents the filling factors in the upper region.

The original text:

“2D mapping of the longitudinal resistance, R_{xx} , versus magnetic field and gate voltage at $T=20\text{mK}$. Data are obtained from the upper region in a quantum well with 0.7nm thick AlAs barrier.”

The new text:

“2D mapping of the longitudinal resistance, R_{xx} , versus magnetic field and gate voltage V_{g1} at $T=20\text{mK}$. Data are obtained from the ohmic contacts along the edge of mesa in the upper region in a quantum well with 0.7nm thick AlAs barrier, with the adjacent region pinched off.”

A similar fan diagram can be obtained in the other regions.

7. In Figs 3a, 3b and 4b, is V or I being continuously tuned? If not, can the authors show discrete data points?

The gate voltages in Figs. 3a & 3c were tuned continuously. In Figs. 3b & 3d we show discrete data points. In Fig. 4b the current, I, was tuned continuously.

8. There is a typo in the caption to figure 3a, where it reads, “ $G_{2T} = (4/3)e^2/h$ to $(2/3)e^2/3$ ”. The last term should be $(2/3)e^2/h$.

The original text:

“In blue area, a $\nu=1$ and a $\nu=1/3$ counter-propagating chiral modes coexist at the interface, and the conductance decreases from $G_{2T}=(4/3)e^2/h$ to $(2/3)e^2/3$ as channel length increases from $6\mu\text{m}$ to $150\mu\text{m}$.”

The new text in the caption of Fig. 3a:

“In the blue area, a $\nu=1$ and a $\nu=1/3$ counter-propagating chiral modes coexist at the interface, and the conductance decreases from $G_{2T}=4e^2/3h$ to $2e^2/3h$ as the channel length increases from $6\mu\text{m}$ to $150\mu\text{m}$.”

9. After mentioning that Refs 13 and 14 predict a crossover between the different $\nu = 2/3$ states, the results of those references are never mentioned again. Is it possible to more specifically state the results of those references and compare them to the experimental results?

Refs. 13 and 14 present a very detailed study of the electric and thermal transport in the $2/3$ state. Results particularly relevant to the present experiment include:

- (i) There is a crossover of the two-terminal conductance from $4e^2/3h$ (the coherent regime) to $2e^2/3h$ (the incoherent regime) with increasing the propagation length L . This crossover is a generic property of a disordered $\nu = 2/3$ edge (*i.e.*, it takes place

for any interaction, tunneling, temperature). It happens when L is around the inter-mode equilibration length ℓ_{eq} .

- (ii) The equilibration length depends on T as a power law; the corresponding exponent is in general non-universal (depending on interaction strength).
- (iii) The conductance approaches $2e^2/3h$ exponentially fast with increasing L/ℓ_{eq} .
- (iv) When the interaction between the modes is so strong and the temperature so low that the system renormalizes to the vicinity of the Kane-Fisher-Polchinski (KFP) fixed point, the conductance shows mesoscopic fluctuations at a broad range of L , followed by the conductance of $2e^2/3h$ in the incoherent regime.
- (v) In the incoherent regime, heat propagates via a diffusive neutral mode.

Comparison to the present experiment:

- (i) Confirmed. The crossover is observed for any value of tunneling. It allows to extract the equilibration length in experiment.
- (ii) The temperature dependence of the equilibration length was not measured in the present experiment but it can clearly be done in future works using this setup.
- (iii) Qualitative agreement. The conductance approaches $2e^2/3h$ very fast with increasing L .
- (iv) For the present experimental parameters, there are no mesoscopic fluctuations, which implies that the system is not close to the KFP fixed point.
- (v) Qualitative agreement. Weakening of the noise with decreasing ℓ_{eq} (or decreasing B) is in agreement with the diffusive propagation of heat.

We have added corresponding clarifications in the Section “Length and magnetic field dependence”:

“These results are consistent with the theoretical prediction [13,14] of a generic (i.e., for any interaction, tunneling, and temperature) crossover in the conductance from $4e^2/3h$ to $2e^2/3h$ with increasing length; the approach to $2e^2/3h$ takes place with exponential accuracy. Further work is needed to accurately study the temperature dependence of the equilibration length, allowing further comparison with theoretical predictions¹³. We also note that the temperature profile along the edge may exhibit interesting structure [14].”

REVIEWERS' COMMENTS:

Reviewer #1 (Remarks to the Author):

The authors have made reasonable attempts to address my concerns and have certainly clarified ambiguities and typos that existed in the original manuscript. Given these changes, I can now recommend publication in Nature Communications.

Reviewer #2 (Remarks to the Author):

The authors have modified their manuscript and added a few schematics in main text and supplementary material to make the device geometry and measurement configuration more transparent.

Among the new figures, Fig.S5 is particularly important, as it provides schematic picture of edge mode configuration along the interface. It is also intended to answer the question raised by reviewer #1 about clarifying "increased overlap in the vertical direction" as well as "increasing the overlapping of the modes in the lateral direction" at lower field.

I want to make sure that I understand this figure correctly. The bottom panel shows (Landau level) energy structure along a linecut in the upper panel in the y direction. For simplicity, Let us define the y value of the interface (where the red and blue levels cross) as y_c . The caption says "edge state energy in the lateral direction", which is confusing because there are no edge states away from the interface. For the blue subband, the $\nu=1$ edge is dismissed for clarity, and this should be confirmed in the caption. Most importantly, it should be mentioned in the caption how the two regimes in the bottom panel ($y < y_c$ and $y > y_c$) correspond to red and blue areas in top panel. For $y < y_c$ in the bottom panel, the $\nu=1$ level in red subband is filled (below fermi level) and the blue $\nu=1/3$ level is empty (above fermi level), so I assume this corresponds to the red area in the top panel, where filling factor is $\nu=1$ for both subbands. It is the opposite for $y > y_c$ where the red $\nu=1$ level is empty (above fermi level) and the blue $\nu=1/3$ level is filled (above fermi level), and filling fraction is $\nu=0$ in the red, and $\nu=4/3$ for the blue subband. This is shown in the following figure, where I marked different y range with the same color code used in the top panel.

If this understanding is correct, there are a couple of issues that should be addressed. First of all, the directions of y -axis are opposite in top and bottom panels. Secondly, the $\nu=1$ ($\nu=1/3$) edge is on the $y < y_c$ ($y > y_c$) side of the interface according to the top panel, therefore the fermi energy should be below the crossing point as shown in the bottom panel. In this case, decreasing fermi energy increases the distance between two edge states and weakens the coupling.

I understand that Fig.S5 is a suggestive plot and the fermi energy could be above the crossing point, in which case two edge states switch positions, the authors interpretation works but the top panel will need to be modified. However, the interpretation depends on the position of fermi surface relative to the crossing point, which appears to be arbitrary without experimental evidence.

Alternatively, my understanding of Fig.S5 could be wrong, in which case I'd be happy to be corrected. At the same time more clarification should be added in figureS5 and its discussions to avoid misleading readers.

After the above mentioned points are address, I'm happy to recommend the paper for publication in Nature Communications.

Reviewer #3 (Remarks to the Author):

The authors have satisfactorily answered my questions and thus I recommend the paper for publication in Nature Communications.

Reviewer 2

We thank Reviewer 2 for his constructive comments! The reply to the Reviewer's comment is presented below.

Reviewer's comments are in black. Our response is in blue. The main changes and additions in the manuscript are in red.

The authors have modified their manuscript and added a few schematics in main text and supplementary material to make the device geometry and measurement configuration more transparent.

Among the new figures, Fig.S5 is particularly important, as it provides schematic picture of edge mode configuration along the interface. It is also intended to answer the question raised by reviewer #1 about clarifying "increased overlap in the vertical direction" as well as "increasing the overlapping of the modes in the lateral direction" at lower field.

I want to make sure that I understand this figure correctly. The bottom panel shows (Landau level) energy structure along a linecut in the upper panel in the y direction. For simplicity, Let us define the y value of the interface (where the red and blue levels cross) as y_c . The caption says "edge state energy in the lateral direction", which is confusing because there are no edge states away from the interface. For the blue subband, the $\nu=1$ edge is dismissed for clarity, and this should be confirmed in the caption. Most importantly, it should be mentioned in the caption how the two regimes in the bottom panel ($y < y_c$ and $y > y_c$) correspond to red and blue areas in top panel. For $y < y_c$ in the bottom panel, the $\nu=1$ level in red subband is filled (below fermi level) and the blue $\nu=1/3$ level is empty (above fermi level), so I assume this corresponds to the red area in the top panel, where filling factor is $\nu=1$ for both subbands. It is the opposite for $y > y_c$ where the red $\nu=1$ level is empty (above fermi level) and the blue $\nu=1/3$ level is filled (above fermi level), and filling fraction is $\nu=0$ in the red, and $\nu=4/3$ for the blue subband. This is shown in the following figure, where I marked different y range with the same color code used in the top panel.

If this understanding is correct, there are a couple of issues that should be addressed. First of all, the directions of y-axis are opposite in top and bottom panels. Secondly, the $\nu=1$ ($\nu=1/3$) edge is on the $y < y_c$ ($y > y_c$) side of the interface according to the top panel, therefore the fermi energy should be below the crossing point as shown in the bottom panel. In this case, decreasing fermi energy increases the distance between two edge states and weakens the coupling.

Indeed Fig. S5 showed inconsistent explanation in its schematic diagrams. We clarify this point by correcting the text and Fig. S5 in the Supplementary Information, but kept the main text unchanged. More specifically:

1. We corrected two points in supplementary Fig. 5b: First, we flipped the red and blue lines but keeping the red $\nu = 1$ and the blue $\nu = 1/3$ in the bottom panel in supplementary Fig. 5, and we colored it to correspond to red and blue regions in the top panel as the Reviewer suggested. Second, we moved the Fermi level below the crossing points.

These changes make the plot consistent with the explanation. When the Fermi level below the crossing point, the $\nu = 1/3$ first (blue) subband is filled in blue region, and the $\nu = 1$ second (red) subband is filled in the red region.

2. We also changed the original caption of Fig. S5.

“A schematic diagram of the edge state’ energies along the y direction”

To the following text

“A schematic diagram of the edge states and Landau Levels energies along the y direction”

3. It should be noted that the $\nu = 1$ first (blue) subband is moving around two regions (so its Landau level is well below the Fermi energy in the interface between the red and the blue regions). We do not show it in the schematic diagram for simplicity.

We thus added “The $\nu = 1$ of the first subband moving around the two regions is dismissed for simplicity” in the caption of Supplementary Figure 5.

4. To better explain the magnetic field dependent coupling, we added the following detailed explanation in Supplementary Note 5.

“As depicted in Supplementary Figure 5 as magnetic field decreases $E_{f1} \rightarrow E_{f2}$, bringing the Fermi Level closer to the ‘crossing point’ of the LLs, consequently, increasing coupling strength between the edges (the lowest field is restricted, as can be seen from the fan diagram)”

We also added the corresponding words in the caption of Supplementary Figure 5:

“By decreasing magnetic field, $E_{f1} \rightarrow E_{f2}$, bringing the Fermi Level closer to the ‘crossing point’ of the LLs, where stronger coupling can be obtained.”

I understand that Fig.S5 is a suggestive plot and the fermi energy could be above the crossing point, in which case two edge states switch positions, the authors interpretation works but the top panel will need to be modified. However, the interpretation depends on the position of fermi surface relative to the crossing point, which appears to be arbitrary without experimental evidence.

Alternatively, my understanding of Fig.S5 could be wrong, in which case I’d be happy to be corrected. At the same time more clarification should be added in figureS5 and its discussions to avoid misleading readers.

After the above mentioned points are address, I’m happy to recommend the paper for publication in Nature Communications.

Note that the Fermi level position relative to the crossing point is not determined arbitrarily, but is rather determined by the value of the magnetic field relative to its value at crossing point - as can be seen in the fan diagram in Fig. 2c. The following suggestive plots illustrate that we are at a magnetic field to the right of the crossing point in the fan diagram (panel a, gate voltage along A_1 to B_1), and the Fermi level is below the crossing point of LLs energy of $\nu = 1$ and $\nu = 1/3$ in panel b.